# The potential value of thromboelastography and conventional coagulation detection in guiding the management of venous thrombosis after traumatic brain injury

Jingchao Zhou[1,2�], Zhenghao Wang[1,2�], Rui Pan[1,2], Pengcheng Zhang[1,2], Hechun Xia[1], Wei Wang[1], Zhanfeng Niu[1,2]*

1 Department of Neurosurgery, General Hospital of Ningxia Medical University, Yinchuan, Ningxia, China,
2 The First Clinical Medical College of Ningxia Medical University, Yinchuan, Ningxia, China

☯ These authors contributed equally to this work.
* niuzfeng228626@163.com

## Abstract

### Background

Coagulation disorders are serious complications of traumatic brain injury (TBI), yet the predictive value of thromboelastography (TEG) for deep vein thrombosis (DVT) in this population remains unclear. This study aimed to assess the TEG profiles and four conventional coagulation parameters in patients with isolated TBI at multiple time points to predict the risk of early DVT.

### Methods

In this retrospective study, 71 patients with isolated TBI were enrolled and categorized into thrombosis (n = 30) and non-thrombosis (n = 41) groups based on Doppler ultrasound and pulmonary angiography findings. Conventional coagulation parameters (PT, APTT, FIB, D-Dimer, TT, PLT) and TEG parameters (R, α-angle, MA) were systematically measured at 24, 72, and 168 hours post-injury. Statistical analyses included group comparisons and binary logistic regression to identify independent predictors of thrombosis.

### Results

Patients who developed thrombosis had significantly lower Glasgow Coma Scale (GCS) scores (p < 0.001). Coagulation profiles evolved dynamically: D-dimer and TEG-MA were elevated at 24 hours; PT, fibrinogen, and D-dimer were higher at 72 hours; PT, D-dimer, and TEG α-angle remained elevated at 168 hours. Multivariate analysis identified lower 24-hour GCS score (OR=0.595, p < 0.001) and elevated PT at 72 and 168 hours as independent predictors of DVT. D-dimer was predictive at 72

**Data availability statement:** All relevant data are within the manuscript and its Supporting Information files.

**Funding:** This work was supported by the Natural Science Foundation of Ningxia Province (Grant No. 2025AAC030802 to Z.N.). The funder had no role in study design, data collection and analysis, decision to publish, or preparation of the manuscript.

**Competing interests:** The authors have declared that no competing interests exist.

hours (OR=1.244, p = 0.023). TEG parameters, despite showing hypercoagulability, did not independently predict thrombosis.

## Conclusions

In isolated TBI, coagulation abnormalities evolved dynamically during the first week after injury. Conventional coagulation parameters, particularly prothrombin time and D-dimer, were more informative for venous thrombosis risk assessment, whereas TEG reflected global hypercoagulability without providing independent predictive value.

## Introduction

Traumatic brain injury (TBI) is a significant global public health concern and represents one of the leading causes of mortality and disability worldwide [1]. Studies indicate that more than 50 million individuals worldwide suffer from TBI annually, with approximately 13.9 million cases occurring in China, representing 18% of the global patient population [2]. Coagulation disorders are a critical factor influencing the clinical progression of TBI. Acute coagulation disorders are among the most clinically significant complications of TBI, potentially leading to secondary brain injury and mortality [3]. Approximately two-thirds of individuals with severe TBI exhibit abnormal findings on routine coagulation tests upon admission to the emergency department [4]. Specifically, TBI can activate the coagulation cascade through the release of brain-derived tissue factors into the systemic circulation, thereby inducing diffuse intravascular coagulation and the formation of cerebral microthrombi. This pathophysiological process is independent of direct hemorrhage. Furthermore, the interplay between thrombosis formation and fibrinolysis during blood coagulation may contribute to an increased risk of delayed or secondary hemorrhage [5,6].

The pathological mechanisms underlying coagulation dysfunction following isolated TBI are complex and remain incompletely understood. Moreover, significant discrepancies persist among the various proposed pathophysiological mechanisms. Coagulation disorders encompass both hypocoagulable states associated with persistent and progressive bleeding [7], and hypercoagulable states characterized by an enhanced predisposition to thrombosis [8]. These two contrasting conditions may coexist following isolated TBI. Coagulation disorders following TBI may lead to hemostatic dysfunction, which significantly increases the risk of further bleeding, organ dysfunction, and adverse clinical outcomes. Therefore, early identification of coagulation disorders is crucial for predicting the development of delayed brain injury and plays a vital role in preventing hemorrhagic complications [9]. During the management of TBI, particularly in severe cases, a secondary hypercoagulable state may develop, markedly elevating the risk of deep vein thrombosis (DVT). The occurrence and detachment of DVT can lead to perioperative pulmonary embolism, which is the third leading cause of cardiovascular disease morbidity and mortality in developed countries [10]. In clinical practice, duplex ultrasound of the lower extremities and

D-dimer (D-D) testing represent the first-line diagnostic modalities for DVT. Routine coagulation parameters are commonly used to assess the patient's coagulation status and guide anticoagulant therapy. However, anticoagulation treatment is frequently associated with severe complications, including life-threatening bleeding and post-discharge recurrence [11]. Therefore, objective assessment of a patient's coagulation status is critically important for the early diagnosis, effective treatment, and prevention of DVT.

Thromboelastography (TEG) is a dynamic monitoring technique that reflects the entire coagulation process, including clot initiation, thrombus formation, and fibrinolysis. It enables comprehensive evaluation of coagulation function and captures real-time changes in platelet (PLT) and fibrin clot formation cascades [12]. Currently, TEG is primarily used in clinical settings for guiding blood transfusions, monitoring coagulation during liver transplantation and cardiovascular surgeries, as well as adjusting anticoagulant and antiplatelet therapies [4]. However, TEG is rarely utilized in patients with TBI, and there is limited evidence supporting its effectiveness in diagnosing or predicting DVT. This study aimed to assess the TEG profiles and four conventional coagulation parameters in patients with isolated TBI at multiple time points to predict the risk of early thrombosis. Based on these predictive indicators, the findings may provide more effective guidance for clinical decision-making and medication management.

## Materials and methods

### Patient population

This retrospective observational study accessed data from patients with TBI who were admitted to the Emergency Department of the Ningxia Medical University Center between January 01, 2021 and February 02, 2022. Data was accessed for study purposes on October 23, 2025. All data were fully anonymized prior to analysis, and the authors had no access to information that could identify individual participants during or after data collection. The study was conducted in accordance with the ethical principles of the Declaration of Helsinki and was approved by the Ethics Committee of the General Hospital of Ningxia Medical University (KYLL-2025–2427). Given the retrospective nature of the study and the use of anonymized data, the requirement for informed consent was waived by the aforementioned ethics committee. The inclusion criteria were as follows: (a) age of >14 years and presentation of isolated TBI, (b) no prior use of anticoagulant or antiplatelet drugs, or other therapeutic agents that may affect coagulation function, (c) clear imaging evidence of TBI was present within 24 hours post-injury. The exclusion criteria were as follows: (a) patients with other cardiovascular and respiratory diseases, severe systemic conditions, sleep disorders, or mental disorders, (b) administration of medications that may affect coagulation function within 7 days post-injury, (c) female patients who experienced their menstrual cycle within 7 days after injury. The exclusion criterion regarding the use of medications affecting coagulation function within 7 days post-injury specifically referred to chronic antiplatelet or anticoagulant therapy prescribed for pre-existing cardiovascular diseases (e.g., aspirin or clopidogrel).

### Data collection and definitions

In this study, an isolated TBI was defined as a computed tomography scan revealing a brain tissue lesion without any other significant accompanying injuries. The Glasgow Coma Scale (GCS) is used to assess the severity of isolated TBI and it is classified into three levels: a score of 13–15 is defined as mild isolated TBI, a score of 9–12 as moderate isolated TBI, and a score of 3–8 as severe isolated TBI. We conducted a comprehensive assessment of the patients, analyzing variables such as the severity of the trauma, age, gender, GCS score at emergency department admission, cause of injury and type of bleeding. Additionally, the clinical necessity of emergency surgery was also evaluated.

Additionally, we collected and analyzed the activated partial thromboplastin time (APTT), fibrinogen (FIB), and prothrombin time (PT) of patients following injury to evaluate alterations in their hemostatic function. This multifaceted approach allows for a comprehensive understanding of the interaction between trauma severity and coagulation parameters, thereby providing clearer insights into the prognosis of emergency patients throughout the clinical course. The GCS

score was independently assessed by two physicians based on the final admission diagnosis and subsequently validated by an independent researcher to ensure the accuracy of the evaluation. Blood samples were collected immediately following injury and at all predetermined time points thereafter. TEG analysis was conducted using the Hematology TEG 5000 Thromboelastography Analyzer (Hematology, Inc., Braintree, MA), with kaolin serving as the activator, to assess citrate-anticoagulated whole blood. The R-time reflects coagulation factor activity, with prolongation indicating coagulation factor deficiency or the presence of anticoagulant agents, and shortening suggesting a hypercoagulable state. The K-time reflects FIB function, with prolongation potentially indicating reduced FIB levels or impaired clot formation, and shortening suggestive of elevated FIB concentration or a hypercoagulable condition. The α angle reflects the rate of fibrin formation, with a decreased angle indicating insufficient FIB or abnormal PLT function. The TEG-MA value reflects the combined contribution of PLT function and FIB to clot strength, with a decrease suggesting thrombocytopenia or functional impairment, and an increase indicating a hypercoagulable state or enhanced PLT activity. Based on standard TEG reference ranges, coagulation status was categorized as follows: a hypercoagulable state was defined by shortened R time and/or elevated α-angle or MA above the upper reference limits; a hypocoagulable state was defined by prolonged R time and/or reduced α-angle or MA below the lower reference limits; patients not meeting these criteria were classified as having normal coagulation status. Coagulation parameters and TEG measurements at 24, 72, and 168 hours were obtained prior to thrombosis diagnosis in all patients.

### Outcomes

The primary outcome of this study was venous thrombosis, defined as the occurrence of DVT and/or pulmonary embolism. DVT was diagnosed using lower-extremity Doppler ultrasound, and pulmonary embolism was confirmed by computed tomography pulmonary angiography. Imaging examinations were performed based on the comprehensive clinical judgment of the attending physician rather than as part of a routine systematic screening protocol. All venous thrombotic events included in this study occurred more than 7 days after injury.

### Peri-injury management

During the study period, venous thromboembolism prevention primarily consisted of mechanical prophylaxis, including intermittent pneumatic compression devices and early mobilization when clinically feasible. Pharmacological thromboprophylaxis was generally not administered within the first 7 days after injury and was considered only after radiological confirmation of intracranial hemorrhage stability.

### Statistical analysis

Data were analyzed using SPSS 27.0 (SPSS Inc., Chicago, IL, USA). For the main analysis, patients were categorized into the thrombosis group and the non-thrombosis group. Clinical characteristics and outcome measurements were classified as continuous or categorical variables based on their nature and were dichotomized using clinically relevant cutoff points. Binary categorical variables were analyzed using the Pearson chi-square test or Fisher's exact test, as appropriate. Continuous variables that followed a normal distribution were analyzed using the independent samples t-test, while non-normally distributed continuous variables were assessed using the Mann-Whitney U test. Correlation analyses were performed using Spearman's rank correlation coefficient to evaluate the associations between coagulation parameters, TEG variables, and thrombosis-related outcomes. These analyses were conducted in the overall study population without stratification. Furthermore, binary regression analysis was conducted to examine the associations among the coagulation index, GCS score, D-D levels, TEG results, and PLT counts. Given the high collinearity observed between the TEG parameters K-time and α-angle, which would compromise model stability, we selected the α-angle as the representative parameter for clot kinetics in the multivariate analysis, and K-time was not included as an independent variable. Statistical significance was defined as a p-value $< 0.05$.

 

## Results

A total of 96 patients were enrolled in this study. Following evaluation, complete clinical data were obtained from 71 patients, comprising 30 in the thrombosis group and 41 in the non-thrombosis group. There were no significant differences between the two groups in terms of age, sex distribution, etiology of injury, or types of intracranial hemorrhage. However, the distribution of GCS scores differed significantly between groups ($p < 0.001$). In the thrombosis group, the majority of patients (63.33%) had severe TBI, defined as a GCS score of 3–8, whereas in the non-thrombosis group, most patients (63.41%) had mild TBI. Furthermore, the proportion of patients who underwent neurosurgical intervention was significantly higher in the thrombosis group (40.00%) than in the non-thrombosis group (17.07%, $p = 0.031$). This indicates that more severe neurological depression may be associated with an increased risk of thrombosis (Table 1). However, when adjusted for other covariates in the multivariate logistic regression model, it was no longer an independent predictor.

Comparative analysis of D-D levels, four coagulation parameters, and PLT counts between the thrombosis and non-thrombosis groups revealed that at the 24-hour time point, D-D levels in the thrombosis group were significantly higher than that in the non-thrombosis group (2.61 ug/mL vs. 1.51 ug/mL, $p = 0.009$). No statistically significant differences were observed in PT, APTT, FIB, TT, or PLT between the two groups at this time point. At the 72-hour time point, the thrombosis group exhibited more pronounced coagulation dysfunction. PT was significantly prolonged ($13.44 \pm 1.52$ s vs. $12.50 \pm 1.06$ s, $p = 0.003$), FIB levels were significantly elevated ($4.09 \pm 1.23$ g/L vs. $3.54 \pm 0.96$ g/L, $p = 0.035$), and D-D levels remained significantly higher compared to the non-thrombosis group (median: 4.03 ug/mL vs. 1.36 ug/mL, $p = 0.008$). No significant intergroup differences were observed in APTT, TT, or PLT. By the 168-hour time point, the difference in PT remained statistically significant, with a higher mean PT in the thrombosis group compared to the non-thrombosis group ($13.20 \pm 1.59$ s vs. $12.30 \pm 1.17$ s, $p = 0.008$). D-D levels also remained persistently elevated in the thrombosis group (1.86

**Table 1. Patient demographic and injury characteristics.**

| Variables | thrombosis | non-thrombosis | p-value |
|---|---|---|---|
| Age (years); mean (SD) | 57.37 (17.24) | 49.95 (18.59) | 0.092 |
| Sex, n (%) | | | 0.964 |
| Male | 25 (83.33%) | 34 (82.93%) | |
| Female | 5 (16.67%) | 7 (17.07%) | |
| GCS scores, n (%) | | | <0.001 |
| 13-15 | 2 (6.67%) | 26 (63.41%) | |
| 9-12 | 9 (30.00%) | 11 (26.83%) | |
| 3-8 | 19 (63.33%) | 4 (9.76%) | |
| Cause of injury, n (%) | | | 0.272 |
| Ground level fall | 10 (33.33%) | 7 (17.07%) | |
| Accident | 16 (53.33%) | 26 (63.41%) | |
| Others | 4 (13.33%) | 8 (19.51%) | |
| Types of bleeding, n (%) | | | 0.228 |
| Subarachnoid | 10 (18.87%) | 9 (21.95%) | |
| Subdural | 12 (22.64%) | 6 (14.63%) | |
| Intraparenchymal | 17 (32.08%) | 8 (19.51%) | |
| Epidural | 14 (26.42%) | 18 (43.90%) | |
| Neurosurgical Procedure, n (%) | 12 (40.00%) | 7 (17.07%) | 0.031 |

Data are presented as mean ± standard deviation (SD) or number (%). Group comparisons were performed using the independent samples t-test for normally distributed continuous variables, and the Pearson chi-square for categorical variables. A p-value < 0.05 was considered statistically significant. GCS: Glasgow Coma Scale.

ug/mL vs. 1.29 ug/mL, p = 0.036). At this time point, the difference in FIB levels was no longer significant, and no statistically significant differences were observed in APTT, TT, or PLT between the two groups (Table 2).

We compared the coagulation function profiles of patients in the thrombosis group and the non-thrombosis group at multiple time points after admission using TEG. The coagulation status was categorized into normal, hypercoagulable, and hypocoagulable states. The results revealed that at the 24-hour time point, there was no significant difference in the distribution of these coagulation states between the two groups (p > 0.05). However, at both the 72-hour and 168-hour time points, the thrombosis group exhibited a significantly higher prevalence of hypercoagulable states compared to the non-thrombosis group (p < 0.001) (Table 3A). Further analysis of individual TEG parameters showed the following (Table 3B): at 24 hours, the TEG-α angle (median: 72.35° vs. 66.60°, p = 0.027) and TEG-MA value (65.58 ± 8.57 mm vs. 59.56 ± 7.01 mm, p = 0.002) in the thrombosis group were significantly higher than those in the non-thrombosis group, while there was no significant difference in TEG-R (p = 0.140). At 72 hours, the TEG-α angle (median: 72.85° vs. 69.50°, p = 0.014) and TEG-MA value (median: 68.55 mm vs. 62.10 mm, p = 0.003) remained significantly higher in the thrombosis group. TEG-R showed no significant difference between the groups (p = 0.078). By the 168-hour time point, TEG-α was significantly higher in the thrombosis group than in the non-thrombosis group (median: 72.55° vs. 70.20°, p = 0.014). TEG-MA was also higher in the thrombosis group but did not reach statistical significance (p = 0.107). TEG-R showed no difference between the groups (p = 0.196).

Based on the observed differences in TEG parameters, four conventional coagulation indicators, and PLT counts between the thrombosis and non-thrombosis groups, as well as comparisons with reference ranges, a comprehensive evaluation of all data was conducted. The "group" (thrombosis vs. non-thrombosis) was defined as the dependent variable, and the following

**Table 2. Laboratory characteristics of thrombosis versus no thrombosis on presentation.**

| Time | Variables | Thrombosis | Non-thrombosis | 95% CI | p-value |
|---|---|---|---|---|---|
| 24h | PT, s | 13.15 ± 1.80 | 12.50 ± 1.40 | −0.13~1.45 | 0.088 |
| | APTT, s | 30.00 ± 4.54 | 29.43 ± 2.34 | −1.23~2.39 | 0.526 |
| | FIB, g/L | 4.03 ± 1.20 | 3.60 ± 1.10 | −0.13~0.98 | 0.124 |
| | D-D, ug/mL | 2.61 (1.41, 8.09) | 1.51(0.64, 3.21) | 0.90~1.13 | 0.009 |
| | TT, s | 14.20 (13.45, 15.68) | 13.80 (13.15, 15.40) | −0.55~1.35 | 0.453 |
| | PLT, × 10⁹/L | 191.93 ± 55.66 | 210.34 ± 62.29 | −46.47~9.66 | 0.203 |
| 72h | PT, s | 13.44 ± 1.52 | 12.50 ± 1.06 | 0.29~1.59 | 0.003 |
| | APTT, s | 29.75 (28.30, 33.35) | 28.80 (26.60, 30.95) | −0.35~2.95 | 0.056 |
| | FIB, g/L | 4.09 ± 1.23 | 3.54 ± 0.96 | 0.02~1.10 | 0.035 |
| | D-D, ug/mL | 4.03 (1.43, 5.59) | 1.36 (0.85, 2.17) | 0.28~3.58 | 0.008 |
| | TT, s | 14.38 ± 1.93 | 14.02 ± 1.78 | −0.54~1.25 | 0.425 |
| | PLT, × 10⁹/L | 202.90 ± 76.75 | 218.32 ± 59.93 | −49.21~18.38 | 0.345 |
| 168h | PT, s | 13.20 ± 1.59 | 12.30 ± 1.17 | 0.24~1.56 | 0.008 |
| | APTT, s | 32.35 (29.03, 34.30) | 30.60 (28.75, 32.30) | −0.20~3.45 | 0.074 |
| | FIB, g/L | 3.65 ± 1.11 | 3.99 ± 1.02 | −0.84~0.16 | 0.183 |
| | D-D, ug/mL | 1.86 (1.26, 3.86) | 1.29 (0.96, 2.02) | 0.02~1.38 | 0.036 |
| | TT, s | 14.30 (13.18, 15.23) | 13.90(12.80, 14.75) | −0.25~0.80 | 0.382 |
| | PLT, × 10⁹/L | 212.00 (144.50, 308.00) | 258.00 (229.50, 287.00) | −101.5 to 6.0 | 0.076 |

Data are presented as mean ± standard deviation or median (interquartile range). Group comparisons were performed using the independent-samples t test or Mann–Whitney U test. The 95% confidence intervals (95% CI) represent between-group differences (thrombosis minus non-thrombosis). Mean differences with Welch-corrected 95% CIs were used for normally distributed variables, and Hodges–Lehmann estimators were used for non-normally distributed variables. APTT: Activated Partial Thromboplastin Time; D-D: D-Dimer; FIB: Fibrinogen; PLT: Platelet Count; PT: Prothrombin Time; TT: Thrombin Time. A p-value < 0.05 was considered statistically significant.

**Table 3. Laboratory characteristics of thrombosis versus no thrombosis on presentation about TEG results.**

**A: Distribution of thromboelastography-defined coagulation states at different time points**

| Time | Thrombosis (Hyper/Hypo/Normal) | Non-thrombosis (Hyper/Hypo/Normal) | p-value |
|------|-------------------------------|------------------------------------|---------|
| 24h | 2/ 2/ 26 | 0/ 3/ 38 | 0.304 |
| 72h | 17/ 1/ 12 | 7/ 1/ 33 | <0.001 |
| 168h | 24/ 2/ 4 | 10/ 4/ 27 | <0.001 |

**B: Comparison of thromboelastography parameters between thrombosis and non-thrombosis groups at different time points**

| Time | Variables | Thrombosis | Non-thrombosis | 95 CI% | p-value |
|------|-----------|-----------|----------------|--------|---------|
| 24h | TEG-R, min | 5.25 (4.40, 6.10) | 4.60 (4.30, 5.30) | −3.20~3.33 | 0.140 |
| | TEG-α, ° | 72.35 (65.40, 74.95) | 66.60 (63.45, 70.20) | −26.20~20.26 | 0.027 |
| | TEG-MA, mm | 65.58±8.57 | 59.56±7.01 | 2.20~9.85 | 0.002 |
| 72h | TEG-R, min | 4.85 (4.68, 5.13) | 4.60 (4.30, 5.20) | −0.26~0.84 | 0.078 |
| | TEG-α, ° | 72.85 (67.20, 75.55) | 69.50 (66.15, 71.80) | −1.59~6.40 | 0.014 |
| | TEG-MA, mm | 68.55 (60.40, 70.23) | 62.10 (58.65, 65.10) | 2.10~7.95 | 0.003 |
| 168h | TEG-R, min | 4.60 (4.18,5.03) | 4.80 (4.40,5.60) | −1.05~0.45 | 0.196 |
| | TEG-α, ° | 72.55 (69.20,75.78) | 70.20 (66.85,72.90) | 0.55~4.80 | 0.014 |
| | TEG-MA, mm | 66.55 (62.40,69.63) | 64.60 (60.40,66.75) | −0.85~4.10 | 0.107 |

Data are presented as mean±standard deviation or median (interquartile range) for continuous variables and counts for categorical variables. Group comparisons were performed using the independent-samples t test or Mann–Whitney U test. The 95% confidence intervals (95% CI) represent between-group differences (thrombosis minus non-thrombosis). Mean differences with Welch-corrected 95% CIs were used for normally distributed variables, and Hodges–Lehmann estimators were used for non-normally distributed variables. TEG-MA: Thromboelastography Maximum Amplitude; TEG-R: Thromboelastography Reaction Time; TEG-α: Thromboelastography Alpha Angle. A p-value<0.05 was considered statistically significant.

parameters were selected as covariates: TEG-R, TEG-α, TEG-MA, PT, PLT counts, and D-D levels. A binary logistic regression analysis was performed. Additionally, the GCS score and neurosurgical procedure was incorporated into the 24-hour model for analysis. The results of the binary logistic regression at different time points were as follows (Table 4): At the 24-hour time point, only the GCS score was a significant predictor of thrombosis (OR = 0.595, p<0.001), whereas all coagulation parameters lacked statistical significance. By 72 hours, prolonged PT emerged as an independent risk factor (OR = 1.867, p=0.010), and elevated D-D levels demonstrated significant predictive value (OR = 1.244, p=0.023). TEG parameters at this time point did not exhibit independent predictive significance. By the 168-hour time point, PT remains a significant independent predictor of thrombosis (OR = 1.729, p=0.013). D-dimer level also approaches statistical significance as a predictive factor (OR = 1.471, p=0.055). Notably, TEG parameters at all three time points did not demonstrate significant predictive value.

In addition, we performed a correlation analysis of TEG parameters with traditional coagulation markers and PLT at multiple time points (Table 5). At 24 hours, TEG-R showed a significant positive correlation with APTT (r=0.397, p=0.001), while TEG-α and TEG-MA were significantly positively correlated with FIB (r=0.378, p=0.001 and r=0.607, p<0.001, respectively). At 72 hours, TEG-R exhibited significant positive correlation with APTT (r=0.306, p=0.010). TEG-α and TEG-MA remained significantly positively correlated with FIB (r=0.380, p=0.001 and r=0.387, p=0.001, respectively). At 168 hours, TEG-R showed significant positive correlation with FIB (r=0.361, p=0.002) and significant negative correlation with D-D (r=−0.249, p=0.036). TEG-MA demonstrated significant positive correlations with FIB (r=0.402, p=0.001) and PLT (r=0.292, p=0.013), and significant negative correlation with TT (r=−0.259, p=0.029). Overall, TEG parameters exhibited dynamically changing correlations with conventional coagulation markers across time points.

## Discussion

Coagulation dysfunction is a well-established predictor of poor clinical outcomes following TBI [13–15]. Despite this recognition, managing these disorders in the perioperative and post-TBI period remains a significant clinical challenge due to a

**Table 4. Binary logistic regression analysis predicting thrombosis by PT, D-D and TEG.**

| Time | Variables | p-value | OR (95%CI) |
|------|-----------|---------|------------|
| 24h | PT, s | 0.774 | 1.088 (0.611–1.939) |
| | D-D, ug/mL | 0.414 | 1.069 (0.910–1.256) |
| | TEG-R, min | 0.890 | 0.938 (0377–2.334) |
| | TEG-α, ° | 0.531 | 0.940 (0.776–1.140) |
| | TEG-MA, mm | 0.190 | 1.118 (0.946–1.322) |
| | GCS | <0.001 | 0.595 (0.441–0.801) |
| | Neurosurgical Procedure | 0.129 | 3.701 (0.682–20.088) |
| 72h | PT, s | 0.010 | 1.867 (1.160–3.004) |
| | D-D, ug/mL | 0.023 | 1.244 (1.030–1.502) |
| | TEG-R, min | 0.211 | 1.803 (0.716–4.541) |
| | TEG-α, ° | 0.472 | 1.049 (0.921–1.194) |
| | TEG-MA, mm | 0.146 | 1.102 (0.967–1.255) |
| 168h | PT, s | 0.013 | 1.729 (1.123–2.661) |
| | D-D, ug/mL | 0.055 | 1.471 (0.991–2.183) |
| | TEG-R, min | 0.972 | 1.012 (0.507–2.022) |
| | TEG-α, ° | 0.347 | 1.098 (0.904–1.334) |
| | TEG-MA, mm | 0.630 | 0.960 (0.811–1.135) |

Data are presented as Odds Ratio (OR), 95% Confidence Interval (95% CI), and p-value. CI: Confidence Interval; D-D: D-Dimer; GCS: Glasgow Coma Scale; OR: Odds Ratio; PT: Prothrombin Time; TEG-R: Thromboelastography Reaction time; TEG-MA: Thromboelastography Maximum Amplitude; TEG-α: Thromboelastography Alpha angle.

lack of targeted strategies. Our study provides a longitudinal analysis of coagulation profiles in patients with isolated TBI, offering insights into the dynamics of thrombotic risk and the comparative value of TEG versus conventional tests.

Our findings align with previous registry studies indicating that patients with isolated TBI and coagulation disorders tend to have lower GCS scores and a higher incidence of thrombotic events [16]. Regression analysis confirmed that a lower GCS score was a significant predictor of thrombosis. However, we posit that this association is likely confounded by injury severity itself. Severe TBI necessitates prolonged immobilization and is intrinsically linked to more profound coagulopathy, both of which are independent risk factors for thrombosis. Accordingly, thrombotic risk following severe TBI reflects the combined impact of trauma severity, prolonged immobilization, and critical care–related interventions. Together, these interacting processes indicate that venous thrombosis in this population represents a multifactorial clinical syndrome rather than the consequence of any single determinant. A similar pattern was observed in patients requiring neurosurgical intervention, further supporting the notion that the disease severity, rather than the GCS score itself, drives the thrombotic risk.

In our cohort, the thrombosis group exhibited significantly elevated PT and D-D levels compared to the non-thrombosis group. This pattern suggests substantial consumption of coagulation factors and rapid fibrin formation and turnover, consistent with systemic activation of the coagulation cascade following brain injury. However, given the observational nature of this study, these findings should be interpreted as descriptive associations rather than evidence of a causal or mechanistic relationship. The proposed pathophysiological interpretations remain speculative. Although prolonged PT and elevated D-D levels may resemble features of disseminated intravascular coagulation, TBI-associated coagulopathy represents a distinct and dynamic pathophysiological process. In the early and subacute phases following TBI, fibrinogen may increase as an acute-phase reactant in response to systemic inflammation and tissue injury. Thus, elevated fibrinogen levels can coexist with markers of coagulation activation and fibrin turnover, reflecting a

**Table 5. Analysis of the correlation between TEG and four coagulation and platelet count.**

| Time | Variables | TEG-R, min | | TEG-α, ° | | TEG-MA, mm | |
|------|-----------|------|---------|------|---------|------|---------|
| | | r | p-value | r | p-value | r | p-value |
| 24h | PT, s | 0.046 | 0.701 | −0.056 | 0.641 | −0.022 | 0.859 |
| | APTT, s | 0.397 | 0.001 | −0.045 | 0.709 | 0.125 | 0.300 |
| | FIB, g/L | −0.078 | 0.516 | 0.378 | 0.001 | 0.607 | <0.001 |
| | D-D, ug/mL | −0.197 | 0.099 | 0.205 | 0.086 | 0.207 | 0.084 |
| | TT, s | 0.029 | 0.809 | 0.128 | 0.288 | 0.166 | 0.168 |
| | PLT, × 10⁹/L | 0.040 | 0.743 | 0.043 | 0.720 | 0.018 | 0.882 |
| 72h | PT, s | 0.194 | 0.105 | 0.058 | 0.633 | 0.084 | 0.486 |
| | APTT, s | 0.306 | 0.010 | 0.047 | 0.697 | 0.175 | 0.144 |
| | FIB, g/L | 0.073 | 0.546 | 0.38 | 0.001 | 0.387 | 0.001 |
| | D-D, ug/mL | −0.195 | 0.103 | 0.159 | 0.185 | 0.234 | 0.050 |
| | TT, s | −0.144 | 0.233 | 0.086 | 0.475 | −0.052 | 0.669 |
| | PLT, × 10⁹/L | −0.195 | 0.103 | 0.067 | 0.577 | 0.025 | 0.837 |
| 168h | PT, s | 0.020 | 0.869 | 0.037 | 0.759 | 0.18 | 0.133 |
| | APTT, s | 0.156 | 0.194 | 0.017 | 0.888 | 0.192 | 0.109 |
| | FIB, g/L | 0.361 | 0.002 | 0.052 | 0.667 | 0.402 | 0.001 |
| | D-D, ug/mL | −0.249 | 0.036 | 0.224 | 0.060 | 0.139 | 0.248 |
| | TT, s | −0.233 | 0.051 | 0.040 | 0.740 | −0.259 | 0.029 |
| | PLT, × 10⁹/L | −0.022 | 0.855 | 0.134 | 0.265 | 0.292 | 0.013 |

Data are presented as Spearman's correlation coefficient (r) and corresponding p-values for each time point. APTT: activated partial thromboplastin time; D-D: D-dimer; FIB: fibrinogen; PT: prothrombin time; PLT: platelet count; TT: thrombin time; TEG-MA: Thromboelastography Maximum Amplitude; TEG-R: Thromboelastography Reaction time; TEG-α: Thromboelastography Alpha angle. A p-value < 0.05 was considered statistically significant.

hypercoagulable and prothrombotic state rather than classical consumptive disseminated intravascular coagulation. The initial hours post-TBI are characterized by coagulopathy, often driven by unregulated thrombin generation, which coincides with the period of highest risk for hemorrhagic progression [17]. The observed hypercoagulable state at 72 and 168 hours, as evidenced by TEG, may result from a complex interplay of factors: trauma-induced hemostatic activation, an impaired fibrinolytic response, and localized intracerebral coagulation from microthrombosis in the pericontusional tissue [18–20]. Isolated TBI can trigger the release of tissue factor into the systemic circulation, promoting a prothrombotic state [5,21]. Conversely, the dynamic imbalance between systemic hypoperfusion, impaired anticoagulant mechanisms, and enhanced fibrinolysis can subsequently lead to consumptive coagulopathy and hemorrhage [5]. Conditions commonly associated with severe TBI, such as hypothermia and acidosis, can further exacerbate hemostatic dysfunction [22,23]. By focusing on patients who did not receive early pharmacological anticoagulation and on thrombosis diagnosed after day 7 post-injury, this study specifically reflects the natural coagulation evolution and untreated thrombosis risk in isolated TBI patients.

Another objective of this study was to evaluate the clinical utility of TEG, a dynamic assessment of hemostasis, compared to conventional coagulation parameters for predicting DVT. Previous studies have demonstrated a significant association between TEG parameters and conventional coagulation indicators [24]. Our TEG analysis revealed a persistent hypercoagulable state in the thrombosis group, characterized by significantly elevated TEG-MA and TEG-α values at 24 hours, and a sustained increase in TEG-α at both 72 and 168 hours post-TBI. Interestingly, despite these clear differences in univariate analysis, none of the TEG parameters retained independent predictive value for thrombosis in the multivariate logistic regression model that included conventional coagulation tests. This finding should be interpreted with caution, as the lack of independent significance may reflect limitations in predictive modeling rather than negating the biological relevance of TEG findings. In the context of a limited sample size and number of outcome events, statistical power may be

insufficient to disentangle the contributions of correlated coagulation variables, thereby attenuating their apparent effects in adjusted models. Consequently, while TEG parameters may not function as independent predictors in this analysis, they remain informative for characterizing global coagulation tendencies in patients with isolated TBI. It should be noted that the use of kaolin-activated thromboelastography primarily reflects clot initiation and strength, providing an integrated assessment of global coagulation dynamics. As such, TEG findings in this study are best interpreted as indicators of overall prothrombotic tendency rather than as a comprehensive evaluation of fibrinolytic activity or platelet inhibition pathways.

Taken together, the hypercoagulability captured by TEG, while present and measurable, may not offer additional independent predictive information for DVT risk beyond what is already provided by conventional parameters in this clinical context. The dynamic and comprehensive nature of TEG makes it well-suited for monitoring the overall trajectory of coagulation status, whereas conventional tests such as PT and D-Dmay possess more specific utility for thrombosis risk stratification in isolated TBI patients.

In contrast, PT proved to be a robust and persistent predictor. Its prolonged elevation at 72 and 168 hours post-injury indicates sustained activation of the extrinsic coagulation pathway and serves as a significant indicator of late thrombotic risk. D-D demonstrated significant predictive value at 72 hours and showed a strong trend towards significance at 168 hours, but its accuracy is likely influenced by trauma-induced hyperfibrinolysis, which affects its specificity. Nevertheless, its consistent elevation in the thrombosis group at early time points underscores its utility as a biomarker for fibrinolysis and early thrombosis screening. The correlation analysis further revealed that the relationships between TEG parameters and conventional markers are not static but evolve over time, reflecting the dynamic nature of post-traumatic coagulopathy.

The management of DVT prophylaxis in isolated TBI is fraught with uncertainty regarding the timing and safety of anticoagulation [25]. Existing evidence indicates that delaying heparin prophylaxis until 24 hours post-injury and restricting its use to patients with a low risk of hemorrhagic progression results in a comparable risk of hematoma expansion to that observed in untreated patients [26]. In our study, we employed TEG in combination with four standard coagulation parameters to dynamically assess patients on days 1, 3, and 7 following TBI. Therefore, for predicting DVT risk, monitoring PT and D-D levels remains crucial in the early phase after TBI. The incorporation of TEG may be best considered when a more comprehensive, real-time assessment of the overall coagulation dynamics is desired, particularly in patients with severe TBI, to guide overall coagulation management rather than solely for DVT prediction. This integrated strategy can enhance the assessment of thrombotic risk and provide a rational basis for initiating anticoagulant therapy, particularly when subsequent neuroimaging confirms the stability of intracranial hemorrhage [27].

## Limitation

Our study has several limitations. First, the single-center, retrospective design with a limited sample size may limit generalizability, preclude severity-stratified analyses, and increase the risk of overfitting. Therefore, our result should be interpreted with cautions. Additionally, institutional practices regarding thromboprophylaxis and imaging indications may have influenced thrombosis detection, and the use of clinically indicated rather than systematic imaging may have introduced detection bias. Finally, the precise onset and peak of coagulation dysfunction following TBI could not be determined, in part because coagulation parameters were assessed at predefined time points (24, 72, and 168 hours), which constrains interpretation of the exact temporal evolution of coagulation abnormalities. Future large-scale, prospective, multicenter studies are warranted to validate these findings and to further clarify the clinical role of TEG and conventional coagulation parameters in thrombosis risk assessment after TBI.

## Conclusions

In patients with isolated TBI, dynamic changes in conventional coagulation parameters and TEG within the first week after injury reflect a sustained hypercoagulable state that may be associated with subsequent venous thrombosis. Patients

who developed thrombosis exhibited persistently elevated D-D levels and modest prolongation of prothrombin time across multiple time points, indicating ongoing activation of the coagulation system during the post-injury period. Although TEG parameters consistently suggested hypercoagulability in the thrombosis group, they did not demonstrate independent predictive value in multivariable analyses, which may be attributable to the limited sample size and number of outcome events. Accordingly, TEG findings should be interpreted as reflecting global coagulation tendencies rather than serving as standalone predictors.

## Supporting information

**S1 Raw Data.** The raw dataset used for statistical analysis is provided as supporting information file.
(XLS)

## Acknowledgments

The authors would like to thank all participants who took part in the study and the graduate student who collaborated with collecting specimens.

## Author contributions

**Conceptualization:** Jingchao Zhou, Zhenghao Wang, Hechun Xia.

**Data curation:** Jingchao Zhou, Zhenghao Wang, Rui Pan, Pengcheng Zhang.

**Formal analysis:** Jingchao Zhou, Zhenghao Wang.

**Funding acquisition:** Zhanfeng Niu.

**Investigation:** Jingchao Zhou, Zhenghao Wang, Rui Pan.

**Methodology:** Jingchao Zhou, Zhenghao Wang, Zhanfeng Niu.

**Project administration:** Hechun Xia, Zhanfeng Niu.

**Resources:** Rui Pan, Pengcheng Zhang, Wei Wang.

**Software:** Pengcheng Zhang.

**Supervision:** Hechun Xia, Wei Wang, Zhanfeng Niu.

**Validation:** Rui Pan, Wei Wang.

**Writing – original draft:** Jingchao Zhou, Zhenghao Wang.

**Writing – review & editing:** Jingchao Zhou, Zhenghao Wang, Zhanfeng Niu.

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
