## [Decision Letter · Decision Letter 0]

29 Dec 2025

Dear Dr. Niu,

Thank you for submitting your manuscript to PLOS ONE. After careful consideration, we feel that it has merit but does not fully meet PLOS ONE’s publication criteria as it currently stands. Therefore, we invite you to submit a revised version of the manuscript that addresses the points raised during the review process.

We look forward to receiving your revised manuscript.

Kind regards,

Tatsuo Shimosawa, M.D., Ph.D.

Academic Editor

PLOS One

Journal Requirements:

The authors would like to thank all participants who took part in the study and the graduate student who collaborated with collecting specimens. This work was supported by the Natural Science Foundation of Ningxia Province (Grant Number: 2025AAC030802).

5. We note you have included a table to which you do not refer in the text of your manuscript. Please ensure that you refer to Table 5 in your text; if accepted, production will need this reference to link the reader to the Table.

Reviewers' comments:

Reviewer's Responses to Questions

**Comments to the Author**

1. Is the manuscript technically sound, and do the data support the conclusions?

Reviewer #1: No

Reviewer #2: Yes

Reviewer #3: Partly

2. Has the statistical analysis been performed appropriately and rigorously?

Reviewer #1: No

Reviewer #2: Yes

Reviewer #3: Yes

3. Have the authors made all data underlying the findings in their manuscript fully available?

Reviewer #1: Yes

Reviewer #2: Yes

Reviewer #3: Yes

4. Is the manuscript presented in an intelligible fashion and written in standard English?

Reviewer #1: Yes

Reviewer #2: Yes

Reviewer #3: No

Reviewer #1: The authors investigated whether thromboelastography (TEG) can help predict venous thrombosis in patients with traumatic brain injury. While the limited sample size restricts the strength of the conclusions, the manuscript provides a relatively detailed longitudinal description of TEG parameters. However, several key methodological details remain insufficiently described—most importantly, the definition of the primary outcome and the clinical protocols for thrombosis prevention, diagnosis, and treatment. My specific comments are as follows.

Methods

1. The rationale for selecting the study period should be clarified, particularly the non-standard end date of February 2.

2. The rationale for including sleep disorders and mental disorders among the exclusion criteria is unclear.

3. Excluding patients who received anticoagulants affecting the coagulation system within 7 days post-injury may have excluded cases in which venous thrombosis developed early, and anticoagulation was initiated during that period. If so, this constitutes an important limitation that should be explicitly acknowledged. In addition, it should be made clear that the results and discussion primarily concern venous thrombosis confirmed after 7 days post-trauma.

4. Because coagulation markers can vary substantially with time after trauma, information on the interval from injury to hospital presentation should be provided.

5. The definition of the primary outcome is unclear. The authors should specify how venous thrombosis was diagnosed, including the diagnostic modality and whether imaging was performed systematically or only when clinically suspected. If a standardized surveillance protocol was used, it should be described.

6. For patients in the thrombosis group, the timing of thrombosis diagnosis should be reported. Interpretation differs significantly depending on whether laboratory/TEG values were obtained before or after thrombosis onset.

7. Given the availability of clinical practice guidelines for the prevention of venous thromboembolism, it is necessary to describe the specific preventive measures implemented in this study.

Results

8. For Tables 1–3, the 95% confidence intervals should be reported in addition to the p-values.

9. For “The coagulation status was categorized into normal, hypercoagulable, and hypocoagulable states.” (Page 12, lines 211-213), definitions for these three categories should be provided in the Methods section.

10. Since K-time is described in the Methods section, corresponding results for K-time should also be presented, or the reason for omission should be stated.

11. With this sample size and number of outcome events, including seven predictors in a multivariable logistic regression model may exceed commonly recommended events-per-variable thresholds.

12. The Methods section does not describe the methods used for correlation analysis. Furthermore, it is unclear whether this was performed in all patients without stratification.

13. Table 5 reports p-values only as thresholds (e.g., p > 0.05) rather than as exact values.

Discussion

14. "In our cohort, the thrombosis group exhibited significantly elevated PT and D-D levels compared to the non-thrombosis group. This pattern suggests substantial consumption of coagulation factors and rapid fibrin formation and turnover, consistent with systemic activation of the coagulation cascade following brain injury." (Page 20, lines 306-309) Given this interpretation, which may suggest a DIC-like pathophysiology, the finding of elevated fibrinogen appears atypical. Further discussion is needed to address this apparent inconsistency.

15. The conclusion states that PT and D-dimer are practical and effective tools for predicting venous thrombosis risk. However, the between-group difference in PT appears to be approximately 1 second, with both values close to the normal range. The clinical application of these findings should be clarified.

16. The reference list appears to require substantial revision.

Reviewer #2: This study examines whether DVT risk can be predicted in patients with head trauma using conventional coagulation parameters and TEG parameters. This is a very interesting, but also very challenging, study. The coagulation parameters used in this study dynamically change over time in patients with head trauma. Furthermore, different patterns are observed depending on the severity of the head injury. Furthermore, thrombus formation, including DVT, affects coagulation parameters. Given the complex interplay of these factors, perhaps a more refined analysis method is needed.

1. When was the DVT diagnosis made? Wouldn't the timing of thrombus formation affect coagulation parameters differently? This factor was not included.

2. Pathological conditions vary considerably depending on the severity of head trauma. Perhaps stratified analysis taking this into account is necessary?

Reviewer #3: 1. Sample Size and Statistical Power

The final analysis includes a relatively small cohort (n = 71), with only 30 thrombosis events. This raises concerns regarding statistical power, particularly for multivariate logistic regression analyses involving multiple covariates. The possibility of model overfitting cannot be excluded, and a formal power calculation is not provided. The authors should acknowledge this limitation more explicitly and interpret regression findings cautiously.

2. Retrospective, Single-Center Design

The retrospective single-center nature of the study introduces potential selection bias and limits external validity. Institutional practices related to thromboprophylaxis, imaging frequency, and ICU management may influence both coagulation parameters and thrombosis rates. This limitation should be emphasized in the Discussion.

3. Lack of Standardization of Thromboprophylaxis

The manuscript does not clearly describe the timing, type, or consistency of pharmacological thromboprophylaxis across patients. Given the strong influence of anticoagulation on both conventional coagulation tests and TEG parameters, this represents a significant confounder. Clarification is required, or this should be addressed as a major limitation.

4. Potential Detection Bias for DVT

DVT diagnosis was based on Doppler ultrasound and pulmonary angiography; however, the frequency and indications for screening are not specified. Patients with more severe TBI or prolonged immobilization may have undergone more frequent imaging, potentially leading to detection bias. This issue should be discussed.

5. Interpretation of TEG Findings

Although TEG parameters demonstrated a persistent hypercoagulable state in the thrombosis group, they did not retain independent predictive value in multivariate analysis. This negative finding may be due to limited sample size rather than true lack of association. The conclusion regarding the limited predictive utility of TEG should therefore be tempered.

6. Incomplete Adjustment for Confounders

Several known risk factors for venous thrombosis—including duration of immobilization, mechanical ventilation, central venous catheterization, transfusion requirements, and infection—were not included in regression models. Residual confounding may influence the reported associations.

7. Causality Cannot Be Established

Given the observational design, causal relationships between coagulation parameters and thrombosis cannot be inferred. The Discussion occasionally implies mechanistic interpretation, which should be clearly framed as speculative.

8. Clarification of TEG Methodology

Only kaolin-activated TEG was used. The authors should acknowledge that this approach may not fully capture fibrinolytic activity or platelet inhibition pathways.

9. Timing of Coagulation Changes

The predefined time points (24, 72, and 168 hours) may miss early or transient coagulation changes immediately post-injury. This limitation should be briefly mentioned.

10. Language and Grammar

Minor grammatical inconsistencies and typographical errors are present throughout the manuscript and should be corrected during revision.

11. Figures and Tables

Some tables are dense and may benefit from simplification or clearer labeling to improve readability.

**Do you want your identity to be public for this peer review?** For information about this choice, including consent withdrawal, please see our Privacy Policy

Reviewer #1: No

Reviewer #2: **Yes:** Eiichi Suehiro

Reviewer #3: No

---

## [Author Response · Author response to Decision Letter 1]

20 Jan 2026

Manuscript ID: PONE-D-25-62755

Title: The Potential Value of Thromboelastography and Conventional Coagulation Detection in Guiding the Management of Venous Thrombosis after Traumatic Brain Injury

Dear Editors and Reviewers,

Thank you for your letter and for the reviewer’s comments concerning our manuscript entitled “The Potential Value of Thromboelastography and Conventional Coagulation Detection in Guiding the Management of Venous Thrombosis after Traumatic Brain Injury” (PONE-D-25-62755). Those comments are all valuable and very helpful for revising and improving our manuscript, as well as the important guiding significance to our researches. We have studied comments carefully and have made correction which we hope will meet with your approval. Revised portion are marked in red in the manuscript. Furthermore, the point-to-point responses have also been presented in this response letter.

We are grateful to you for allowing us to revise our manuscript and look forward to hearing from you soon.

Best wishes

Sincerely,

Zhanfeng Niu

Response to Academic Editor and Journal Requirements

Comment 1: PLOS ONE style and file naming requirements

Response: We have carefully revised the manuscript to comply with PLOS ONE formatting and style requirements in accordance with the provided templates. Specifically, we changed the format of the title to be written in sentence case. And all the main sections have been given first-level headings.

Comment 2: Inconsistency in grant information

Response: Thank you for pointing this out. We have reviewed the information provided in the ‘Funding Information’ and ‘Financial Disclosure’ sections and have ensured that they now match and are consistent with each other.

Comment 3: Funding information incorrectly included in the Acknowledgments section

Response: We thank the editor for this clarification. All funding-related text has been removed from the Acknowledgments section of the manuscript. Funding information is now reported exclusively in the Funding Statement via the online submission form. We have included the amended Funding Statement in the cover letter and kindly request that the submission system be updated accordingly.

Comment 4: Ethics statement location

Response: We have revised the manuscript so that the ethics statement appears only in the Methods section, and any duplicate statements in other sections have been removed.

Comment 4: Missing in-text reference to Table 5

Response: Thank you for noting this. We have now added an explicit in-text reference to Table 5 in the Results section to ensure proper linkage. (page 17, line 295)

Comment 6: Suggested citations

Response: We confirm that the reviewer comments did not include specific recommendations to cite additional previously published works. Therefore, no changes were required in this regard.

Responds to the reviewer’s comments:

Reviewer #1:

1. The rationale for selecting the study period should be clarified, particularly the non-standard end date of February 2.

Response: We appreciate the reviewer's point. The study period endpoint (February 2, 2022) was determined by the need to maintain consistency in core measurement data. This study focuses on the comparison of detailed coagulation parameters, including thromboelastography (TEG) indices, across different time points and patient groups. The TEG equipment used in our institution underwent a scheduled upgrade in February 2022, which could have affected absolute values and reference ranges. To avoid potential bias related to differences in measurement systems and to ensure comparability of coagulation data across patients, only individuals who completed all required testing prior to the equipment upgrade were included in the analysis. As a result, February 2, 2022 represents the natural endpoint of the study period.

2. The rationale for including sleep disorders and mental disorders among the exclusion criteria is unclear.

Response: Thank you to the reviewers for pointing this out. Regarding the exclusion of patients with mental disorders, this decision was primarily based on careful consideration of controlling for confounding medication factors and ensuring data reliability. For patients with mental disorders, neither the patients themselves nor their family members could accurately provide a complete history of long-term oral medications that might affect coagulation function. Therefore, to ensure the quality of the data upon which our analysis relies and to avoid drawing biased conclusions due to uncontrolled confounding medication factors, we decided to exclude these patients from the study cohort.

The exclusion of patients with sleep disorders, such as those with obstructive sleep apnea (OSA), is supported by substantial evidence demonstrating that OSA is an independent risk factor for deep vein thrombosis and pulmonary embolism[1, 2]. Including such patients would prevent us from clearly discerning to what extent the observed abnormalities in coagulation parameters and thrombotic events in our study could be attributed to acute TBI itself, leading to this exclusion decision.

[1] PENG Y-H, LIAO W-C, CHUNG W-S, et al. Association between obstructive sleep apnea and deep vein thrombosis / pulmonary embolism: A population-based retrospective cohort study [J]. Thrombosis Research, 2014, 134(2): 340-5.

[2] ZHANG J, GU J, KUANG Y, et al. Prevalence of obstructive sleep apnea in venous thromboembolism: a systematic review and meta-analysis [J]. Sleep and Breathing, 2019, 23(4): 1283-9.

3. Excluding patients who received anticoagulants affecting the coagulation system within 7 days post-injury may have excluded cases in which venous thrombosis developed early, and anticoagulation was initiated during that period. If so, this constitutes an important limitation that should be explicitly acknowledged. In addition, it should be made clear that the results and discussion primarily concern venous thrombosis confirmed after 7 days post-trauma.

Response: We thank the reviewer for raising this important concern. The exclusion criterion stating “use of medications that may affect coagulation function within 7 days post-injury” was intended to exclude patients receiving chronic antiplatelet or anticoagulant therapy (such as aspirin or clopidogrel) for pre-existing cardiovascular diseases.

The timing of pharmacological anticoagulation in patients with TBI varies considerably in clinical practice, ranging from as early as 72 hours to several weeks after injury. However, considering the risk of hemorrhagic complications, in our institution, pharmacological anticoagulation is usually initiated no earlier than 7 days after injury and only after radiological confirmation of intracranial hemorrhage stability. Therefore, initiation of anticoagulation within the first 7 days after injury is uncommon in our center.

The primary objective of this study was to characterize the natural evolution of coagulation profiles and the risk of thrombosis in patients with isolated TBI who did not receive early pharmacological anticoagulation. Initiation of oral or systemic anticoagulant therapy within 7 days post-injury would inevitably interfere with the spontaneous process of thrombus formation, rendering such patients unrepresentative of an untreated TBI-associated thrombosis risk population.

Furthermore, all venous thrombotic events included in this study were diagnosed after day 7 post-injury using lower-extremity Doppler ultrasound or computed tomography pulmonary angiography (CTPA). We have now clarified this point in the Methods section and emphasized the importance of this time window for interpretation of our findings in the Discussion.

To address this comment and improve methodological clarity, we have made the following revisions to the manuscript:

a. Methods – Patient population (page 6, lines 119-122)

We have clarified the definition of anticoagulant exposure within 7 days post-injury.

Added text:

The exclusion criterion regarding the use of medications affecting coagulation function within 7 days post-injury specifically referred to chronic antiplatelet or anticoagulant therapy prescribed for pre-existing cardiovascular diseases (e.g., aspirin or clopidogrel).

b. Methods – Outcomes (page 8, lines 167-168)

We have explicitly reported the timing and diagnostic modalities of venous thrombosis.

Added text:

All venous thrombotic events included in this study occurred more than 7 days after injury.

c. Discussion (page 21, lines 358-360)

We have emphasized the study population and the interpretation of findings in relation to the predefined time window.

Added text:

By focusing on patients who did not receive early pharmacological anticoagulation and on thrombosis diagnosed after day 7 post-injury, this study specifically reflects the natural coagulation evolution and untreated thrombosis risk in isolated TBI patients.

4. Because coagulation markers can vary substantially with time after trauma, information on the interval from injury to hospital presentation should be provided.

Response: We appreciate the reviewers' suggestions. This study focuses on systemic coagulation changes at predefined time points of approximately 24, 72, and 168 hours post-injury, rather than the transient fluctuations in the extremely early stage (< 24 hours). All patients were tested at the 24-hour time point after injury, ensuring the consistency of the time points. Although there were differences in the specific time from injury to admission, the focus of this study is on the longitudinal coagulation trend within one week after injury, rather than the absolute values at the moment of admission. Therefore, we believe that the current design can effectively capture the evolution pattern of thrombosis risk related to clinical practice and provide a basis for early warning.

5. The definition of the primary outcome is unclear. The authors should specify how venous thrombosis was diagnosed, including the diagnostic modality and whether imaging was performed systematically or only when clinically suspected. If a standardized surveillance protocol was used, it should be described.

Response: We thank the reviewers for pointing out this key issue. In this study, venous thrombosis was defined as deep vein thrombosis (DVT) and/or pulmonary embolism confirmed by objective imaging modalities. DVT was diagnosed using lower-extremity Doppler ultrasound, and pulmonary embolism was confirmed by computed tomography pulmonary angiography.

Imaging examinations were not performed as part of a routine systematic screening protocol for all patients. Instead, Doppler ultrasound and/or computed tomography pulmonary angiography were performed based on comprehensive clinical judgment, including limb swelling, unexplained hypoxemia, sudden deterioration in oxygenation, or elevated thrombotic risk as judged by the treating physicians.

To improve clarity, we have revised the Methods section to explicitly define the primary outcome and describe the diagnostic modalities and indications for imaging. To improve clarity and avoid redundancy, we have revised the Methods section by consolidating all information related to outcome definition, diagnostic modalities, and imaging indications into the Outcomes subsection. Accordingly, the original sentence describing thrombus detection has been removed, as these details are now explicitly and comprehensively defined in the Outcomes section.

Methods – Outcomes (page 8, lines 162-167)

Added text:

The primary outcome of this study was venous thrombosis, defined as the occurrence of DVT and/or pulmonary embolism. DVT was diagnosed using lower-extremity Doppler ultrasound, and pulmonary embolism was confirmed by computed tomography pulmonary angiography. Imaging examinations were performed based on the comprehensive clinical judgment of the attending physician rather than as part of a routine systematic screening protocol.

6. For patients in the thrombosis group, the timing of thrombosis diagnosis should be reported. Interpretation differs significantly depending on whether laboratory/TEG values were obtained before or after thrombosis onset.

Response: We thank the reviewer for this important comment. In the present study, all venous thrombotic events were diagnosed after day 7 post-injury, using lower-extremity Doppler ultrasound and/or computed tomography pulmonary angiography. Importantly, all laboratory coagulation parameters and TEG measurements at 24, 72, and 168 hours were obtained prior to thrombosis diagnosis.

Therefore, the coagulation and TEG values analyzed in this study reflect coagulation changes preceding thrombosis detection, rather than alterations secondary to the thrombotic event itself or to anticoagulant therapy initiated after diagnosis. This temporal relationship supports the interpretation of these parameters as potential early indicators of thrombosis risk.

To improve clarity, we removed a previously existing sentence describing coagulation and TEG measurements at 24, 72, and 168 hours post-injury and we have explicitly reported the timing of thrombosis diagnosis and its relationship to laboratory and TEG measurements in the Methods section.

Methods –Data collection and definitions (page 8, lines 158-160)

Added text:

Coagulation parameters and TEG measurements at 24, 72, and 168 hours were obtained prior to thrombosis diagnosis in all patients.

Methods – Outcomes (page 8, lines 167-168)

Added text:

All venous thrombotic events included in this study occurred more than 7 days after injury.

7. Given the availability of clinical practice guidelines for the prevention of venous thromboembolism, it is necessary to describe the specific preventive measures implemented in this study.

Response: We thank the reviewer for this important comment. During the study period, venous thromboembolism prevention in patients with isolated traumatic brain injury primarily consisted of mechanical prophylaxis, including intermittent pneumatic compression devices and early mobilization when clinically feasible. Pharmacological thromboprophylaxis was generally not initiated during the first 7 days after injury, due to concerns regarding hemorrhagic progression, and was considered only after radiological confirmation of intracranial hemorrhage stability.

Thus, the study cohort largely represents isolated TBI patients managed without early pharmacological anticoagulation, allowing assessment of coagulation dynamics and thrombosis risk in the early post-injury phase.

To improve clarity and transparency, we have added a description of VTE preventive measures to the Methods section.

Methods – Peri-injury management (page 9, lines 170-174)

Added text:

During the study period, venous thromboembolism prevention primarily consisted of mechanical prophylaxis, including intermittent pneumatic compression devices and early mobilization when clinically feasible. Pharmacological thromboprophylaxis was generally not administered within the first 7 days after injury and was considered only after radiological confirmation of intracranial hemorrhage stability.

8. For Tables 1–3, the 95% confidence intervals should be reported in addition to the p-values.

Response: Thank you for this valuable suggestion. We have added 95% confidence intervals (95% CI) for all between-group comparisons in Tables 2 and 3. No 95% CI was added to Table 1, as it presents baseline characteristics without inferential comparisons.

9. For “The coagulation status was categorized into normal, hypercoagulable, and hypocoagulable states.” (Page 12, lines 211-213), definitions for these three categories should be provided in the Methods section.

Response: We thank the reviewer for this important comment. We have now explicitly defined the criteria used to classify coagulation status as normal, hypercoagulable, or hypocoagulable based on standard TEG reference ranges.

Methods – Data collection and definitions (page 8, lines 153-158)

Added text:

Based on standard TEG reference ranges, coagulation status was categorized as follows: a hypercoagulable state was defined by shortened R time and/or elevated α-angle or MA above the upper reference limits; a hypocoagulable state was defined by prolonged R time and/or reduced α-angle or MA

---

## [Decision Letter · Decision Letter 1]

1 Feb 2026

The potential value of thromboelastography and conventional coagulation detection in guiding the management of venous thrombosis after traumatic brain injury

PONE-D-25-62755R1

Dear Dr. Niu,

We’re pleased to inform you that your manuscript has been judged scientifically suitable for publication and will be formally accepted for publication once it meets all outstanding technical requirements.

Kind regards,

Tatsuo Shimosawa, M.D., Ph.D.

Academic Editor

PLOS One

Additional Editor Comments (optional):

Reviewers' comments:

Reviewer's Responses to Questions

**Comments to the Author**

Reviewer #1: All comments have been addressed

Reviewer #2: All comments have been addressed

Reviewer #3: All comments have been addressed

2. Is the manuscript technically sound, and do the data support the conclusions?

Reviewer #1: (No Response)

Reviewer #2: Yes

Reviewer #3: Yes

3. Has the statistical analysis been performed appropriately and rigorously?

Reviewer #1: (No Response)

Reviewer #2: Yes

Reviewer #3: Yes

4. Have the authors made all data underlying the findings in their manuscript fully available?

Reviewer #1: (No Response)

Reviewer #2: Yes

Reviewer #3: Yes

5. Is the manuscript presented in an intelligible fashion and written in standard English?

Reviewer #1: (No Response)

Reviewer #2: Yes

Reviewer #3: Yes

Reviewer #1: I thank the authors for their careful and comprehensive responses. All of my concerns have been addressed.

Reviewer #2: Thank you for resubmitting and your honest response to my comment. I think this paper will be useful to readers.

Reviewer #3: The authors have responded thoroughly and professionally. Most issues are appropriately acknowledged, and limitations are clearly integrated into the manuscript. The tone is measured, and overinterpretation has largely been corrected. That said, several responses rely heavily on post-hoc framing rather than methodological mitigation, which should be noted.

The authors have substantively engaged with my critiques, strengthened the manuscript’s methodological transparency, and appropriately tempered interpretation. Remaining limitations are largely intrinsic to the study design and are now clearly acknowledged. No major unresolved issues remain.

**Do you want your identity to be public for this peer review?** For information about this choice, including consent withdrawal, please see our Privacy Policy

Reviewer #1: No

Reviewer #2: **Yes:** Eiichi Suehiro

Reviewer #3: **Yes:** Dr. Ishwar Singh

---

## [Editor Report · Acceptance letter]

PONE-D-25-62755R1

PLOS One

Dear Dr. Niu,

I'm pleased to inform you that your manuscript has been deemed suitable for publication in PLOS One. Congratulations! Your manuscript is now being handed over to our production team.

Kind regards,

on behalf of

Prof. Tatsuo Shimosawa

Academic Editor

PLOS One